# MULTI-AGENT GAME GENERATION AND EVALUATION VIA AUDIO-VISUAL RECORDINGS

## ABSTRACT

Generating novel video games is a challenging problem. Large Language Models (LLMs) can generate games and animations, but lack automated evaluation metrics and struggle with complex content. To tackle these issues, we built a new metric and multi-agent system. First, we propose AVR-Eval, a metric for multimedia content where a model compares the Audio-Visual Recordings (AVRs) of two contents and determines which one is better. We show that AVR-Eval properly identifies good from broken or mismatched content. Second, we built AVR-Agent, a multi-agent system to generate JavaScript code from a bank of multimedia assets (audio, images, 3D models) and using AVR feedback. We show higher AVR-Eval with AVR-Agent than one-shot prompt. However, while humans benefit from high-quality assets and audio-visual feedback, they do not significantly increase AVR-Eval for LLMs. This reveals a gap between humans and AI content creation.

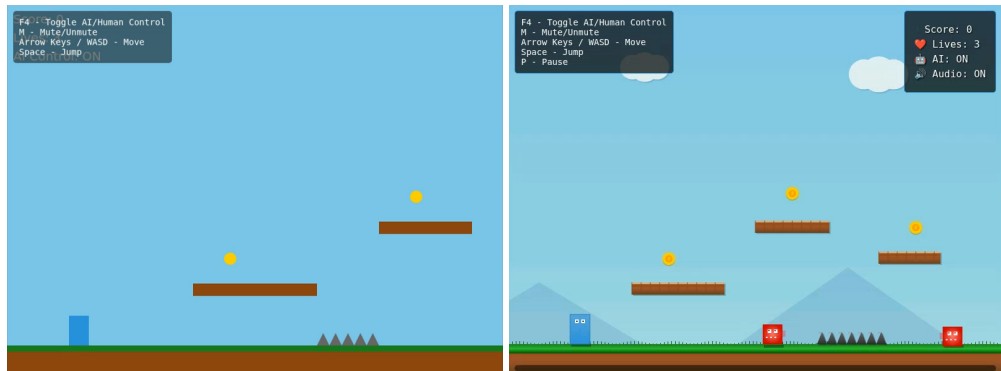

Figure 1: Platformer game generation with Kimi-K2: single prompt vs AVR-Agent (10 iterations).

## 1 INTRODUCTION

Using AI to generate interactive multimedia content (such as video games and animations) is challenging. By definition, multimedia uses a wide range of content modalities. For example, a JavaScript or Flash game can contain images (with or without transparency), vector graphics, videos, 3D models, music, sounds, fonts, code, and much more. Many of these modalities are not naturally handled by current large pretrained models. Furthermore, the interactive aspect involves human interaction through a mouse and keyboard. Currently, there are two main lines of direction for generating interactive multimedia: 1) controllable audio and video generation, and 2) coding assistants/agents.

The first approach, Controllable audio and video generation, focuses on replicating the audio-visual output with controls. This approach is agnostic to the medium; it does not need access to the source of the content (code or file). Recent progress has been made for video generation with audio (Google DeepMind, 2025) and controllable video generation without audio (Menapace et al., 2021; Yang et al., 2024; Valevski et al., 2024; Che et al., 2024; Yu et al., 2025; Kanervisto et al., 2025). We are not aware of any method for controllable video with audio generation in the context of multimedia

content creation, such as video games. However, it is just a matter of time before this arises. While generating video with audio can work well for animations, it is not ideal for video games given their limited context length. For example, in a first-person view, looking behind and waiting long enough before looking forward will cause the front scene to completely change due to lost memory. Some works are working on the memory problem (Xiao et al., 2025; Wu et al., 2025), but this is a very challenging, non-trivial problem. Furthermore, a games involve a polished experience with a beginning and end, which is not currently the case with controllable video generation.

The second approach, coding assistants/agents, focuses on generating the code for, say, an HTML page with a JavaScript video game or animation. Large Language Models (LLMs) have been used to generate such code directly in one-shot or as AI assistants (Hu et al., 2024; Anjum et al., 2024; Rosebud AI, 2024; X, 2025; Wang et al., 2025; Anysphere). However, one-shot generation is not enough to build complex, fully fleshed-out content; it takes humans years of work and access to artists' assets in order to finish one good game. Instead of one-shot generation, AI assistants rely on back-and-forth interactions between the LLM and a human, coming with their own assets to develop their own video game. While powerful, it is not the AI making the game. There exist many fully autonomous agent systems for coding (Hong et al., 2023; Chen et al., 2023; Tufano et al., 2024; Wang et al., 2024); however, most of these methods are limited to text and potentially vision, but not audio and not involving assets of various modalities.

Regarding the evaluation of interactive multimedia content. FVD (Unterthiner et al., 2019) can be used to assess closeness in video to the true video distribution. However, this requires a real-world dataset of such content, and handling audio is more complicated. In practice, we want to generate new content for which there is no available dataset. For web content, the main approach is WebDev Arena, a specific version of Chatbot Arena (Chiang et al., 2024), which compares side-by-side web content implementations of the same prompt by two models. It relies on human evaluation; a human decides which content is better. We would like a similar metric but fully automated, similar to coding benchmarks with LLM-as-a-Judge (Zheng et al., 2023; Li et al., 2024).

In this work, we focus on the web coding agent direction. More specifically, our goal is *fully autonomous generation and evaluation of interactive multimedia content containing audio*.

Our contributions:

1. We leverage Audio-Visual Recordings (AVR) to construct a relative evaluation metric (choosing the best content out of 2 contents) for multimedia content (AVR-Eval) using text and omni-modal models as judges.

2. We build a multi-agent framework for JavaScript multimedia content generation through a bank of multimedia assets (images, audio, 3D models), audio-visual recording feedback, and console logs (AVR-Agent).

3. We test AVR-Agent on a set of games and animations showing that current state-of-the-art coding models benefit from AVR-Agent, but, contrary to humans, struggle at leveraging multimedia assets and audio-visual feedback for benefit.

## 2 METHOD

### 2.1 EVALUATION METRIC BASED ON AUDIO-VISUAL RECORDINGS (AVR-EVAL)

To assess the quality of the generated games and animations, we use the following criteria:

- Description Fidelity: How well does the {content-type} match the following description? Description: {content-description}
- Visual Design: How appealing are the graphics and animations? Are colors, shapes, and layout harmonious?
- Audio Quality: How well does the audio (sound effects and music) align with the content and enhance its quality?
- Behavior Correctness: Are there any broken behaviors?

For video games, the following criteria are added:

- Gameplay Quality: How engaging and fun is the gameplay?
- AI Player Quality: How well does the AI play the game?

For animations, the following criteria are added:

- Smoothness: How smooth and fluid are the animations? Are key frames and timing polished?
- Creativity and Originality: How creative and interesting is the animation?

Given these criteria, we seek to obtain an evaluation metric that favors working content over i) broken (e.g., title-screen only, black screen) and ii) mislabeled (e.g., fireworks animation when the goal is a bouncing-ball animation) content. In addition, we would like the evaluation to favor better content over worse content; this is quite subjective, but one way to push in that direction is to iii) favor human-made over generated content.

To achieve that goal, AVR-Eval (Figure 2) compares two contents (A and B) to decide which is better. An Audio-Visual Recording is made for each content. Then, we provide a *multiround* prompt to an omni-modal model as follows: prompt1) describe content A (given video and audio), prompt2) describe content B (given video and audio), and prompt3) given the criteria, determine which content (A or B) is better? Then, a stronger text model is asked to review the omni-modal evaluation and ultimately decide which content is truly the best based on the criteria. We use Qwen2.5-Omni-7B (Xu et al., 2025) for the omni-modal model and Qwen3-32B (Yang et al., 2025) for the text model.

We provide an ablation (Table 1) showing that removing either the multiround (by directly comparing both content in a single prompt with two videos and two audios), relative comparison (by evaluating one content at a time instead of the two together), or the review (by directly extracting the best content from the omni-modal model response) worsen reliability against broken and mislabeled contents. We attribute this to the following: 1) multiround: the descriptions may ground the model reducing hallucinations and the model is probably undertrained on multiple videos and audios within a single prompt, 2) relative: relative comparisons may ground the choice and the model is likely untrained for evaluating video games, and 3) review: state-of-the-art methods for omni-modal understanding (e.g., Qwen2.5-Omni-7B) are still not as good on reasoning and instruction following as state-of-the-art text models.

For all experiments, we compare both sides: A vs B and B vs A with temperature 0. For i), we compare 5 working animations (1 of each type; see Table 2) to 12 broken animations, for a total of 120 comparisons. For ii), we compare 5 working animations (1 of each type; see Table 2) to 8-10 animations of other (incorrect) types (e.g., working "fireworks" content compared to "bouncing-ball" content, but the description is "fireworks"), for a total of 92 comparisons. For iii), we compare 9 working generated platformer video games to 5 high-quality human-made platformer video games, for a total of 90 comparisons. The generated games are AI-controlled, directly implemented in the JavaScript code, and the human games are controlled by a human player.

We show that AVR-Eval rarely chooses broken (0.91%) or mislabeled JavaScript content (6.47%) over working JavaScript content. It also prefers human-made high-quality content over generated content 67.78% of the time. Currently, AVR uses Qwen2.5-Omni-7B; as better omni-modal models become available, we expect AVR to improve in alignment with human preferences.

Table 1: Ablation for the AVR evaluation metric: Mean (standard deviation) win rate when comparing working generated content versus broken, mislabeled, or human-made content.

| Evaluation metric | | | | % Win against | | |
| | | | | Objective | | Subjective |
| | Multiround | Relative | Review | Broken ↑ | Mislabeled ↑ | Human-made ↓ |
|---|---|---|---|---|---|---|
| AVR | ✓ | ✓ | ✓ | **99.09** (2.03) | **93.53** (4.42) | 32.22 (12.01) |
| | ✗ | ✓ | ✓ | 90.00 (9.85) | 80.02 (17.09) | **28.89** (16.91) |
| | ✗ | ✗ | ✓ | 90.91 (8.50) | 81.39 (16.99) | **28.89** (16.91) |
| | ✗ | ✓ | ✗ | 9.09 (6.43) | 3.25 (4.64) | 73.33 (16.58) |
| | ✗ | ✗ | ✗ | 9.09 (8.50) | 6.25 (8.20) | 77.78 (13.94) |

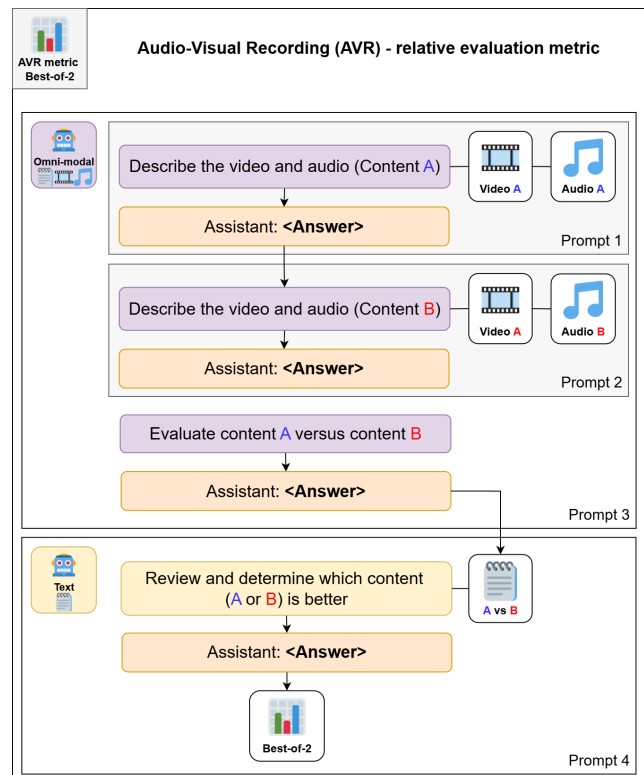

Figure 2: Audio-Visual Recording Evaluation metric (AVR-Eval)

## 2.2 SIMPLE MULTIMEDIA GENERATION BENCHMARK

With AVR-Eval, we can compare different generated content to determine which generative method is better. We leverage this metric to build a simple set of 5 video games and 5 animations to be generated (Table 2). We make it very simple and open-handed to assess the creativity of the model.

Table 2: Easy-Moderate Benchmark. Simple animations (easy) and games (moderate difficulty).

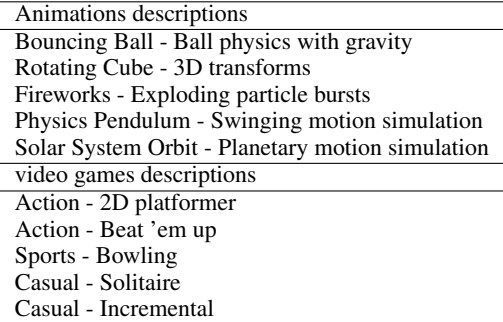

| Animations descriptions |
| --- |
| Bouncing Ball - Ball physics with gravity |
| Rotating Cube - 3D transforms |
| Fireworks - Exploding particle bursts |
| Physics Pendulum - Swinging motion simulation |
| Solar System Orbit - Planetary motion simulation |
| video games descriptions |
| Action - 2D platformer |
| Action - Beat 'em up |
| Sports - Bowling |
| Casual - Solitaire |
| Casual - Incremental |

## 2.3 MULTI-AGENT FRAMEWORK FOR AUDIO-VISUAL CONTENT GENERATION (AVR-AGENT)

Now that we have a validated evaluation metric and a simple benchmark, we can test different generative models at the task of generating video games and animations. While some content can be generated in 1-shot by extremely large ($\geq$ 500B parameters) and powerful coding LLMs, the reality is that smaller open-source LLMs (e.g., Qwen3-32B) will often generate broken content, and complex content may take multiple iterations of improvements to succeed or reach a certain quality, even with strong models. To leverage multiple improvement steps, multimedia assets, and Audio-

Visual Feedback (AVR), we build our own custom agent framework specifically designed for the creation of multimedia JavaScript content.

Figure 3: AVR-Agent: Multi-agent framework for audio-visual content generation

The AVR-Agent framework (Figure 3) uses two agents: a text-only coding model (we use many different models, see the results section below) with an omni-modal model (Qwen2.5-Omni-7B). We accumulated a bank of asset packs from itch.io and kenney.nl with permissive licenses. The asset packs contain images, audio (music and sounds), 3D models (.glb) made specifically for video-game creation. See Appendix C for the full list.

In the first stage, the coding model selects which assets to use to produce the desired content given the original description. We give it a maximum of 50 samples from 5 sample packs. In theory, one could use Retrieval-Augmented Generation (RAG) to select content, but to our knowledge, there is no RAG adapted to work with any multimedia content (images, audio without speech, and 3D objects). Our approach is to simply select the 5 packs based on their names and content (the number of each file type in the pack). Then, in each pack, we provide the names and details of the assets (BPM and duration for audio; dimensions for images; animation names for 3D models), and the coding agent must choose its samples. The end result is a small directory tree showing the 50 assets the model will have access to.

In the second stage, the coding model is asked to generate the content based on the original description, chosen assets, general guidelines, and evaluation criteria. In practice, we found the initial content to have enormous influence on the future evolution of the content, and since smaller models (Qwen3-32B) can often generate broken or bad content, we found it helpful to generate $k$ candidate

initial contents and leverage the AVR-Eval metric to determine the best-out-of-$k$ initial candidate to keep. T

In the third stage, we improve the content over multiple steps. At each step, the content is rendered in a browser, and an Audio-Visual Recording (AVR) is made. The console logs (containing errors and warnings) are extracted during rendering, and the AVR is fed to the omni-modal model to produce AVR Feedback by asking the model to i) describe the content, and ii) provide subjective feedback about the content based on the evaluation criteria. Then, the coding model is asked to improve the content given the base information (the original description, chosen assets, criteria, guidelines), current code, AVR Feedback, and console logs.

By default, due to autoplay policies (Google, 2017), audio is disabled in web browsers and interaction (clicking) is necessary to enable audio. Thus, in the guidelines, we asked the model to make a start button with a specific ID, and we had the browser automatically interact with the button.

To handle games, for the Audio-Visual Recordings (AVR) to be meaningful, we cannot just start the content since the player is human-controlled and will not move automatically. To handle this, we added as a guideline that the agent must implement AI automated controls by default (and that they can be disabled by pressing F4 for human control).

Given this setup, we show below that we improve on the generated content (compared to one-shot) generation.

## 3 EXPERIMENTS

### 3.1 SETUP

We test our multi-agent framework on the 5 games and animations of the Benchmark (see Table 2) with 9 coding models consisting of 2 closed-source models: Gemini-2.5-Flash, Grok-3-Mini, 3 large open-source models Kimi-K2-1T (Moonshot AI, 2025), Qwen3-Coder-480B, DeepSeek-v3-0324-671B, and 4 small models: Devstral-Small-2505-24B (Mistral AI, 2025), Qwen3-32B (Yang et al., 2025), Qwen2.5-Coder-32B (Hui et al., 2024), DeepSeek-Coder-V2-Lite-16B (Guo et al., 2024).

For closed-source and large open-source models, we use 10 improvement iterations or 5 improvement iterations with best-of-5 initial contents, given the API cost. For other models, which run on our hardware (4 GPUs), we use 20 improvement iterations or 10 improvement iterations and best-of-10 initial contents. Note that if there is any console log error at the end, we allow up to 2 extra improvement steps to resolve the error in order to prevent an error made at the last improvement step from affecting the final content. We always use Qwen-Omni-7B (Xu et al., 2025) as the omni-modal model.

We run 3 different sets of experiments to compare the performance a) for the same model across different settings, b) between initial and final content, and c) for the same setting across different models.

a) We seek to determine how well the coding agent can leverage the assets, the audio-visual feedback, and the Best-of-$k$ initial content (Table 3). To do so, we compare the final content with or without i) audio-visual feedback, ii) assets[1], and iii) Best-of-$k$ for the initial content generation. Thus, 8 methods are compared to 8-1=7 methods with two orderings (A vs B and B vs A) for a total of 14 comparisons across each method per content. In total, there are 10*9*8*(8-1)*2=10080 comparisons (Dataset a).

b) We seek to determine the benefits of the agent framework. To do so, we compare the initial content to the final content after improvement steps (Table 4). Since we use two orderings (A vs B and B vs A), there are 2 comparisons per method and content. In total, there are 10*9*8*2=1440 comparisons (Dataset b).

c) We seek to determine which coding model is the best (Table 5). To do so, we compare models against each other for the same setting. Thus, 9 models are compared to 9-1=8 models with two orderings (A vs B and B vs A) for a total of 8 comparisons per method and contents. In total, there are 10*9*8*8*2=11520 comparisons (Dataset c).

---

[1]Note that audio can still be generated without assets through Tone.js by making synthetic sounds.

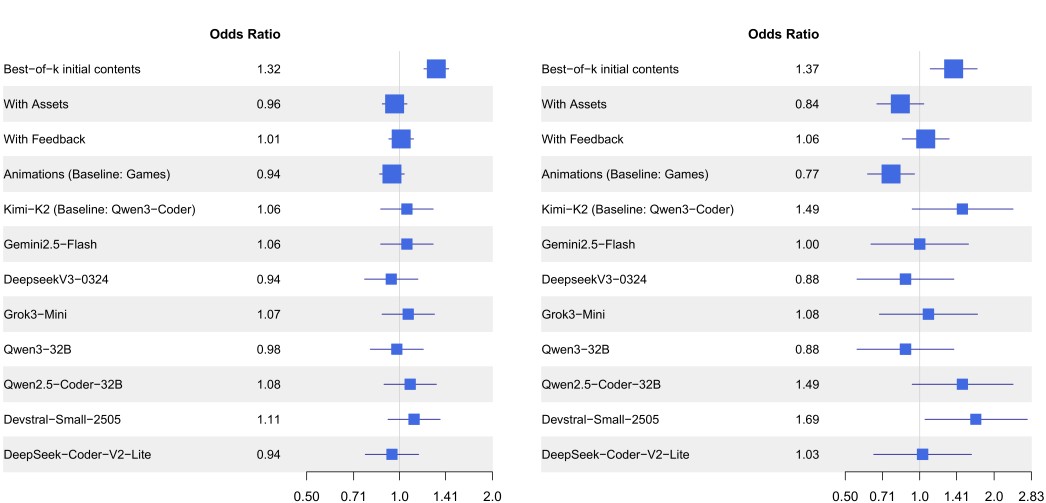

(a) Predicting the win rate of generated content against different settings (with and without feedback, assets, best-of-k initial contents) based on $n = 10080$ samples. Best-of-$k$ significantly increase win rate against other settings.

(b) Predicting the win rate of generated content of final generated content against initial (one-shot) content based on $n = 1440$ samples. Best-of-$k$ and Devstral-Small-2505 have significantly higher win rate, while animations, Animations have significantly lower win rate rate against initial content.

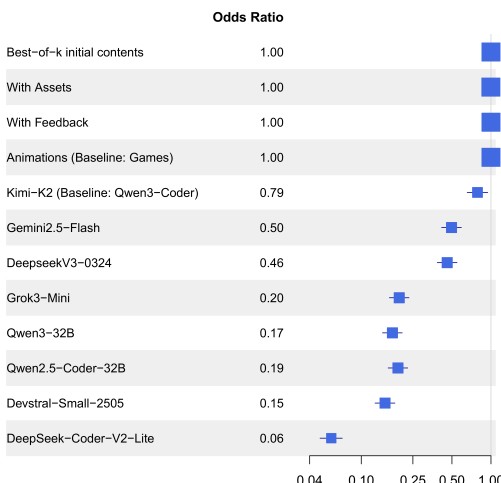

(c) Predicting the win rate of generated content against different models based on $n = 11520$ samples. Qwen3-Coder has significantly higher win rate against other models.

Figure 4: Logistic regression on relative evaluations between different contents with AVR-Eval

Using the three datasets formed from those experiments, we fit logistic regression models to predict the probability of winning conditional on the following one-hot features: with assets, with feedback, with Best-of-$k$ initial content, animations (relative to games as baseline), model used (relative to Kimi-K2 as baseline).

## 3.2 RESULTS

Logistic regression results are shown in Figure 4. We also provide results in raw form through the percentage win rate per setting and model in Tables 3, 4, and 5 of Appendix A.

In a), using Best-of-$k$ initial contents significantly increases the win rate against other settings, showing that it is beneficial. Furthermore, the settings with the best win rate for each model always contained Best-of-$k$ initial contents (see Table 3). However, there was no significant effect from including assets or feedback.

In b), using Best-of-$k$ initial contents and Devstral-Small-2505 significantly improves win rate against initial content, while animations are significantly detrimental (since animations are easier, there is less benefit from improvement steps). We also fit a logistic regression with no features (bias-only) and find that the base probability of the final content winning over initial content is $0.647$, $95\%$ CI $[0.622, 0.671]$, which means that AVR-Agent has a significant positive effect over one-shot generation. In concordance with this finding, we found that $79.2\%$ of the settings with AVR-Agent had a higher win rate than one-shot generation, and $100\%$ of the best settings with AVR-Agent (from Table 3) had higher win rate than one-shot generation (see Table 4).

In c), Qwen3-Coder-480B has a significantly higher win rate against other models, followed closely by Kimi-K2-T1. We observed similar findings (Qwen3-Coder-480B and Kimi-K2-T1 having higher win rates) in Table 3. We provide screenshots of the generated games by Kimi-K2-T1 in Section E.

Overall, this shows that AVR-agent significantly improves the quality of the content over one-shot generation, and choosing the best-out-of-$k$ initial content is generally better than extending generation with $k$ additional iterations. However, performance does not generally improve from providing high-quality human-made assets, and adding audio-visual feedback from an omni-modal model to the coding model. Furthermore, Qwen3-Coder and Kimi-K2 are the strongest coding models in the list of models we tested.

## 4 CONCLUSION

We proposed AVR-Eval, an evaluation metric for multimedia content through Audio-Visual Recording (AVR). AVR-Eval compares two implementations of the same content. Through AVR-Eval, new coding models will be able to evaluate and improve their models against other existing models in an automated fashion without requiring human evaluation.

While coding models can sometimes produce good JavaScript content in one shot, they are not always perfect. To tackle this issue, we built a multi-agent framework for HTML JavaScript multimedia content generation (AVR-Agent) by leveraging AVR for choosing the best initial content, console logs, audio-visual feedback, and a bank of multimedia assets (images, audio, 3D models).

We tested AVR-Agent on a small set of 5 games and 5 animations using AVR-Eval as a metric. AVR-Agent was beneficial over one-shot generation, and choosing the best initial content out of $k$ candidates was better than training with $k$ additional iterations. However, the coding agent did not benefit from using assets and receiving audio-visual feedback. Thus, while humans significantly benefit from having pre-made high-quality assets and require audio-visual feedback to debug and improve games, it appears that current coding models do not leverage those effectively. This shows a gap between humans and machines.

Regarding the lack of benefit of assets, we suspect that current models are trained to work without separate assets or using placeholders; hence, they are not trained to leverage real assets. To use assets properly, it would help if the coding model could process them directly as images and audio in the input prompt. Regarding the lack of benefit from the omni-modal feedback, it could be that text feedback is insufficient, or again, an artifact of their pretraining not relying on it. Once omni-modal models are strong enough for coding, we expect them to be able to leverage multimedia assets and

audio-visual feedback for improved performance (see Section D). Sadly, the current state-of-the-art Qwen-2.5-Omni-7B model is incapable of coding at a reasonable performance; we found it unable to generate any working content, hence why we did not use it for coding.

Note that AVR-Eval is not yet perfect (e.g., it chooses broken content 0.91% of the time over working content), and it has not been directly tested on human preference. We expect future omni-modal models to improve on these aspects since AVR-Eval can be used with any omni-modal model that can process audio (including music and sounds; many models only handle speech), videos, and text.

Overall, AVR-Eval and AVR-Agent are a first step toward automated game design, but to truly achieve impressive feats of game design using these techniques, we need strong omni-modal models that can code well.

Note that models that could not fit within our hardware (4 GPUs) were only tested with few iterations (either 5 initial and 5 improvements or 1 initial and 10 improvements) due to the API cost, and we did not test the more expensive state-of-the-art closed-source models. Everything was paid out-of-pocket, hence the limitations in how much we could test. We tested smaller models that could fit within our 4 available GPUs. However, even when they are considered state-of-the-art, smaller models (e.g., Qwen3-32B) tend to produce broken games. This shows that we still have a long way to go for reliable small pretrained models on coding tasks.

We would like to acknowledge the art content made by domi.wav (Dominic Sandefur), David KBD, TomMusic (Thomas Devlin), Yogi (Tronimal), OmegaPixelArt, doranarasi, and Kenney that was used in the asset bank. See Appendix C for more details.

ETHICS STATEMENT

Generating images, videos, and video games with AI is controversial, as it generally produces worse content and removes work from artists. This work is focused on assessing the current model's ability to generate video games, but we acknowledge that the use of AI for such cases can cause harm to real-world artists and could cause a deluge of low-quality AI-generated games.

In this paper, we used a bank of human-made multimedia assets (images, videos, sounds, music). We made sure to credit every artist in both the paper (see Appendix C) and the code. We also made sure to only use assets with open licenses that do not prohibit their artistic content from being used for AI-generated content.

REPRODUCIBILITY STATEMENT

The links to all the data (multimedia assets) are provided in the appendix and in the code. The code to reproduce all experiments is provided. However, reproducing this work for closed-source models requires spending money on API credits.

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

# A   ADDITIONAL RESULTS

In the manuscript, we analyzed the results through logistic regression models to predict the win rates. Here we show the win rates per setting. This gives a more direct overview of the results. We report the mean and standard deviation over the 5+5=10 contents.

Table 3: Mean (standard deviation) win rate of generated content against different settings (with and without feedback, assets, best-of-k initial contents) over 5 games and 5 animations.

| % Win against other settings | | | | | | | |
|---|---|---|---|---|---|---|---|
| Without assets | | | | With assets | | | |
| Without feedback | | With feedback | | Without feedback | | With feedback | |
| ∅ | Init-best | ∅ | Init-best | ∅ | Init-best | ∅ | Init-best |
| *Qwen3-Coder-480B* | | | | | | | |
| 47.9 (21.0) | 47.1 (25.7) | 39.3 (18.5) | 55.7 (16.1) | 44.3 (27.3) | 52.9 (21.3) | 52.1 (17.8) | **60.0** (14.8) |
| *Kimi-K2-1T* | | | | | | | |
| 35.7 (17.5) | 49.3 (26.4) | 36.4 (22.2) | 52.1 (23.8) | 58.6 (19.3) | **65.0** (17.6) | 60.7 (19.1) | 42.1 (21.7) |
| *Gemini-2.5-Flash (Closed-source)* | | | | | | | |
| 60.0 (21.6) | 51.4 (14.2) | 62.1 (19.9) | 55.7 (13.4) | 26.4 (22.3) | **64.3** (17.5) | 37.9 (32.1) | 41.4 (23.3) |
| *Grok-3-Mini (Closed-source)* | | | | | | | |
| 50.0 (16.1) | 62.9 (26.7) | 51.4 (25.6) | 52.9 (36.6) | 23.6 (23.8) | 48.6 (24.9) | 37.1 (27.3) | **73.6** (16.5) |
| *DeepSeek-v3-0324-671B* | | | | | | | |
| 52.1 (17.8) | **69.3** (27.4) | 46.4 (24.3) | 54.3 (19.1) | 40.0 (23.6) | 58.6 (27.5) | 26.4 (22.8) | 52.1 (26.9) |
| *Devstral-Small-2505-24B* | | | | | | | |
| 42.9 (26.5) | 39.3 (28.0) | 45.0 (21.8) | **66.4** (21.0) | 47.9 (23.8) | 45.7 (31.4) | 50.0 (22.3) | 62.9 (24.0) |
| *Qwen3-32B* | | | | | | | |
| 41.4 (28.9) | **64.3** (23.8) | 47.1 (30.9) | 59.3 (31.2) | 56.4 (24.2) | 42.9 (27.1) | 37.1 (29.3) | 51.4 (25.8) |
| *Qwen2.5-Coder-32B* | | | | | | | |
| 56.4 (26.8) | **63.6** (27.7) | 49.3 (28.7) | 50.0 (25.4) | 42.9 (30.7) | 40.7 (28.0) | 38.6 (22.9) | 58.6 (24.2) |
| *DeepSeek-Coder-V2-Lite-16B* | | | | | | | |
| 47.9 (19.9) | 48.6 (31.9) | 41.4 (33.1) | 57.1 (31.0) | 55.0 (24.1) | 40.7 (29.4) | 48.6 (27.7) | **60.0** (34.5) |

% of times where *Init-best* is better than ∅: **27/36=75%** overall; **9/9=100%** in the **best settings**
% of times where *With assets* is better than *Without assets*: **16/36=44.4%** overall; **5/9=55.5%** in the **best settings**
% of times where *With feedback* is better than *Without feedback*: **21/36=58.3%** overall; **3/9=33.3%** in the **best settings**

Table 4: Mean (standard deviation) win rate of final generated content against initial (one-shot) content over 5 games and 5 animations at different settings. A win rate above $50\%$ indicates a benefit from the multi-agent framework over one-shot generation.

| % Win against initial (one-shot) content | | | | | | | |
|---|---|---|---|---|---|---|---|
| Without assets | | | | With assets | | | |
| Without feedback | | With feedback | | Without feedback | | With feedback | |
| ∅ | Init-best | ∅ | Init-best | ∅ | Init-best | ∅ | Init-best |
| *Qwen3-Coder-480B* | | | | | | | |
| **75.0** (42.5) | **40.0** (45.9) | 50.0 (40.8) | 65.0 (33.7) | 70.0 (42.2) | 50.0 (52.7) | **75.0** (35.4) | 70.0 (35.0) |
| *Kimi-K2-1T* | | | | | | | |
| 75.0 (26.4) | **85.0** (24.2) | 50.0 (40.8) | 75.0 (42.5) | 60.0 (31.6) | **85.0** (33.7) | **85.0** (24.2) | 50.0 (47.1) |
| *Gemini-2.5-Flash (Closed-source)* | | | | | | | |
| 70.0 (35.0) | 70.0 (35.0) | 75.0 (35.4) | **45.0** (36.9) | 50.0 (40.8) | **80.0** (35.0) | **40.0** (39.4) | 65.0 (33.7) |
| *Grok-3-Mini (Closed-source)* | | | | | | | |
| 60.0 (39.4) | **85.0** (33.7) | 70.0 (35.0) | 60.0 (39.4) | 55.0 (43.8) | 70.0 (42.2) | **40.0** (45.9) | 70.0 (35) |
| *DeepSeek-v3-0324-671B* | | | | | | | |
| 60.0 (39.4) | 55.0 (43.8) | **70.0** (35.0) | 65.0 (24.2) | 55.0 (43.8) | 55.0 (43.8) | **45.0** (43.8) | 65.0 (41.2) |
| *Devstral-Small-2505-24B* | | | | | | | |
| 55.0 (43.8) | 75.0 (42.5) | 75.0 (35.4) | 75.0 (42.5) | 70.0 (35.0) | 75.0 (42.5) | 75.0 (26.4) | **85.0** (24.2) |
| *Qwen3-32B* | | | | | | | |
| 65.0 (41.2) | **70.0** (35.0) | **70.0** (35.0) | 60.0 (45.9) | 50.0 (40.8) | **35.0** (41.2) | 50.0 (47.1) | **70.0** (25.8) |
| *Qwen2.5-Coder-32B* | | | | | | | |
| 65.0 (41.2) | **85.0** (24.2) | 70.0 (42.2) | 80.0 (35.0) | 55.0 (43.8) | 60.0 (39.4) | 70.0 (35.0) | 80.0 (35.0) |
| *DeepSeek-Coder-V2-Lite-16B* | | | | | | | |
| 60.0 (45.9) | 60.0 (45.9) | 50.0 (40.8) | **85.0** (24.2) | 50.0 (40.8) | 70.0 (42.2) | **40.0** (39.4) | **85.0** (24.2) |

% of times where *win rate ¿ 50%*: **57/72=79.2%** overall; **9/9=100%** in the **best settings** of Table 3

**Results**   In Table 4, we find that AVR-Agent generally improves the quality of the content over one-shot generation (79.5% of the time better overall and 100% with the best settings). Further-

Table 5: Mean (standard deviation) win rate of generated content against other models over 5 games and 5 animations at different settings. Highest win rate within a column indicates the best model.

| % Win against other models | | | | | | | |
|---|---|---|---|---|---|---|---|
| Without assets | | | | With assets | | | |
| Without feedback | | With feedback | | Without feedback | | With feedback | |
| ∅ | Init-best | ∅ | Init-best | ∅ | Init-best | ∅ | Init-best |
| *Qwen3-Coder-480B* | | | | | | | |
| **75.0** (16.7) | **83.1** (15.3) | 71.2 (24.0) | **84.4** (12.2) | 77.5 (11.5) | 70.6 (24.1) | 81.2 (12.5) | **81.2** (13.2) |
| *Kimi-K2-1T* | | | | | | | |
| **75.0** (16.7) | 65.6 (13.9) | 60.0 (27.4) | 75.0 (13.2) | **82.5** (14.1) | **81.2** (11.0) | **81.9** (11.2) | 68.1 (18.5) |
| *Gemini-2.5-Flash (Closed-source)* | | | | | | | |
| 65.6 (23.8) | 70.0 (13.8) | **73.8** (23.2) | 64.4 (15.9) | 46.9 (27.7) | 73.8 (10.5) | 56.9 (28.5) | 59.4 (22.7) |
| *Grok-3-Mini (Closed-source)* | | | | | | | |
| 40.0 (23.2) | 43.1 (20.9) | 40.6 (20.0) | 36.2 (28.1) | 28.7 (22.1) | 40.0 (25.5) | 40.6 (28.9) | 59.4 (15.4) |
| *DeepSeek-v3-0324-671B* | | | | | | | |
| 65.6 (21.1) | 66.9 (23.2) | 69.4 (10.0) | 61.9 (13.0) | 62.5 (16.1) | 66.2 (25.2) | 48.8 (23.0) | 55.0 (29.9) |
| *Devstral-Small-2505-24B* | | | | | | | |
| 29.4 (16.7) | 26.2 (18.8) | 35.6 (19.8) | 30.0 (15.8) | 43.1 (17) | 35.6 (16.7) | 42.5 (16.6) | 38.8 (20.2) |
| *Qwen3-32B* | | | | | | | |
| 33.1 (28.4) | 42.5 (17.6) | 37.5 (18.9) | 44.4 (27.9) | 44.4 (26.8) | 32.5 (20.4) | 36.2 (28.7) | 35.0 (18.0) |
| *Qwen2.5-Coder-32B* | | | | | | | |
| 47.5 (19.6) | 43.8 (19.1) | 44.4 (28.9) | 33.8 (22.1) | 42.5 (29.0) | 33.8 (21.9) | 40.6 (18.0) | 38.1 (29.8) |
| *DeepSeek-Coder-V2-Lite-16B* | | | | | | | |
| 18.8 (16.4) | 8.8 (7.9) | 17.5 (18.1) | 20.0 (16.1) | 21.2 (20.0) | 16.2 (19.6) | 21.2 (22.5) | 14.4 (14.7) |

more, in Table 4, we find that choosing the best-out-of-$k$ initial content is generally better than extending generation with $k$ additional iterations (75% of the time better overall and 100% with the best settings).

However, in Table 4, we observe that performance does not generally improve from pre-selecting assets from a bank of high-quality human-made assets and providing them to the coding model (44.4% of the time better overall, but 55.5% with the best settings). Similarly, performance did not generally improve from adding audio-visual feedback from an omni-modal model to the coding model (58.3% of the time better overall and 33.3% with the best settings).

Regarding the different models, we found Qwen3-Coder-480B and Kimi-K2-1T to be generally better than other models.

These results concord with the results of the logistic regression analyses from Figure 4.

## B    AVR-AGENT GUIDELINES

In AVR-Agent, the agents receive the following guidelines.

The base instructions are:

- Be contained in a single HTML file.
- You can use HTML5 Canvas and any javascript library via CDN (e.g., Phaser, Three.js, PixiJS, Babylon.js, Matter.js).
- Assume that the user does not have a GPU; the code should run well on CPUs.
- Have clear, well-commented code with meaningful variable names.
- Implement smooth animations for all moving elements.
- Include a title screen with a large button that has id='start-button'. Pressing 'enter' or clicking the button should press the button and start the {content-type}. Ensure that audio only starts after pressing the start button.
- DO NOT use alerts (e.g., alert("Game Over!"))

The video-game specific instructions are:

- Include AI to control the player by default; it should play the game in a smart way.

- Allow switching to human control when F4 is pressed.
- Include game state management and responsive control.
- No broken behaviors (softlock, hardlock, hitbox bugs, clipping, AI breakdown, etc.).
- Use clear, visually distinct elements for game objects.
- Ensure visual feedback for player actions and game events.
- Use appropriate colors and visual effects to enhance gameplay.
- Maintain consistent visual style throughout the game.
- Include background music that fits the theme and mood of the game.
- Add sound effects for key game events (jumps, collisions, item collection).
- Implement audio controls (mute/unmute) with the 'M' key".
- Ensure audio volume is balanced and not overwhelming.

The animation specific instructions are:

- Include interesting visual elements and transitions.
- Focus on aesthetic appeal.
- Respect physical laws if relevant to the requested animation.
- No broken behaviors (jank, broken keyframes, hitbox bugs, clipping, etc.).
- Create visually appealing elements with attention to detail.
- Implement appropriate visual effects to enhance the animation.
- Ensure consistent visual style throughout the animation.
- Use color and composition effectively to convey mood and theme.
- Include background music that complements the animation's mood and pace.
- Add sound effects for key animation events and transitions.
- Implement audio controls (mute/unmute) with the 'M' key".
- Synchronize audio timing with visual elements.

## C  AVR-AGENT ASSETS

The AVR-Agent uses a bank of asset packs with permissive licenses. They are described in Table 6.

Table 6: List of asset packs used is the AVR-Agent

| Asset pack | Author | License |
|---|---|---|
| 8 Bit RPG Adventure and Fantasy Music Pack | domi.wav (Dominic Sandefur) | None |
| belmont-chronicles-metroidvania-music-pack | David KBD | CC By 4.0 |
| cosmic-journey-space-themed-music-pack | David KBD | CC By 4.0 |
| eternity-metal-scfi-music-pack | David KBD | CC By 4.0 |
| Free-Fantasy-SFX-Pack-By-TomMusic | TomMusic (Thomas Devlin) | No resale, redistribution |
| Free-Game-Boy-Music-Pack | Yogi (Tronimal) | None |
| gameboy-sfx-pack | OmegaPixelArt | None |
| hexapuppies-synthwave-music-pack | David KBD | CC By 4.0 |
| interstellar-edm-metal-music-pack | David KBD | CC By 4.0 |
| pink-bloom-synthwave-music-pack | David KBD | CC By 4.0 |
| SHMUP-MIDI-Pack ogg&m4a | doranarasi | No resale, redistribution, NFT |
| Kenney assets | Kenney | CC0 |

Regarding the Kenney assets, we use the following asset packs: kenney-boardgame-pack, kenney-car-kit, kenney-casino-audio, kenney-castle-kit, kenney-city-kit-commercial-20, kenney-city-kit-roads, kenney-city-kit-suburban-20, kenney-cursor-pack, kenney-digital-audio, kenney-fish-pack-2, kenney-food-kit, kenney-holiday-kit, kenney-impact-sounds, kenney-interface-sounds, kenney-jumper-pack, kenney-letter-tiles, kenney-mini-arcade, kenney-mini-arena, kenney-mini-characters, kenney-mini-dungeon, kenney-minigolf-kit, kenney-music-jingles, kenney-new-platformer-pack-1.0, kenney-pirate-kit, kenney-pirate-pack, kenney-pixel-vehicle-pack, kenney-planets, kenney-platformer-kit, kenney-playing-cards-pack, kenney-puzzle-pack, kenney-puzzle-pack-2, kenney-racing-pack, kenney-rpg-audio, kenney-sci-fi-sounds, kenney-scribble-dungeons, kenney-scribble-platformer, kenney-shooting-gallery, kenney-simple-space, kenney-sketch-town-expansion, kenney-space-shooter-extension, kenney-space-station-kit, kenney-sports-pack, kenney-tanks, kenney-toon-characters-1, kenney-top-down-tanks-redux, kenney-tower-defense, kenney-tower-defense-kit, kenney-toy-brick-pack, kenney-toy-car-kit, kenney-ui-audio, kenney-ui-pack, kenney-voiceover-pack, kenney-voiceover-pack-fighter, kenney-voxel-pack, kenney-watercraft-pack, kenneymedals, kenney-background-elements, kenney-background-elements-redux, kenney-blaster-kit, kenney-block-pack, kenney-blocky-characters-20, kenney-board-game-icons.

## D AVR-AGENT 2.0 (FOR AN OMNI-MODEL AGENT CAPABLE OF CODING)

We currently rely on AVR text feedback after processing by an omni-modal agent. This is needed because the coding agent cannot process Audio Visual Recording (AVR) directly. Current omni-modal models are not strong enough for coding, so they cannot be used as such. In the future, there will be omni-modal models with strong coding capabilities that will be able to directly process the AVR. We show in Figure 5 what the AVR-Agent would look like in this case. The current code already implements this feature, which will be useful in the future.

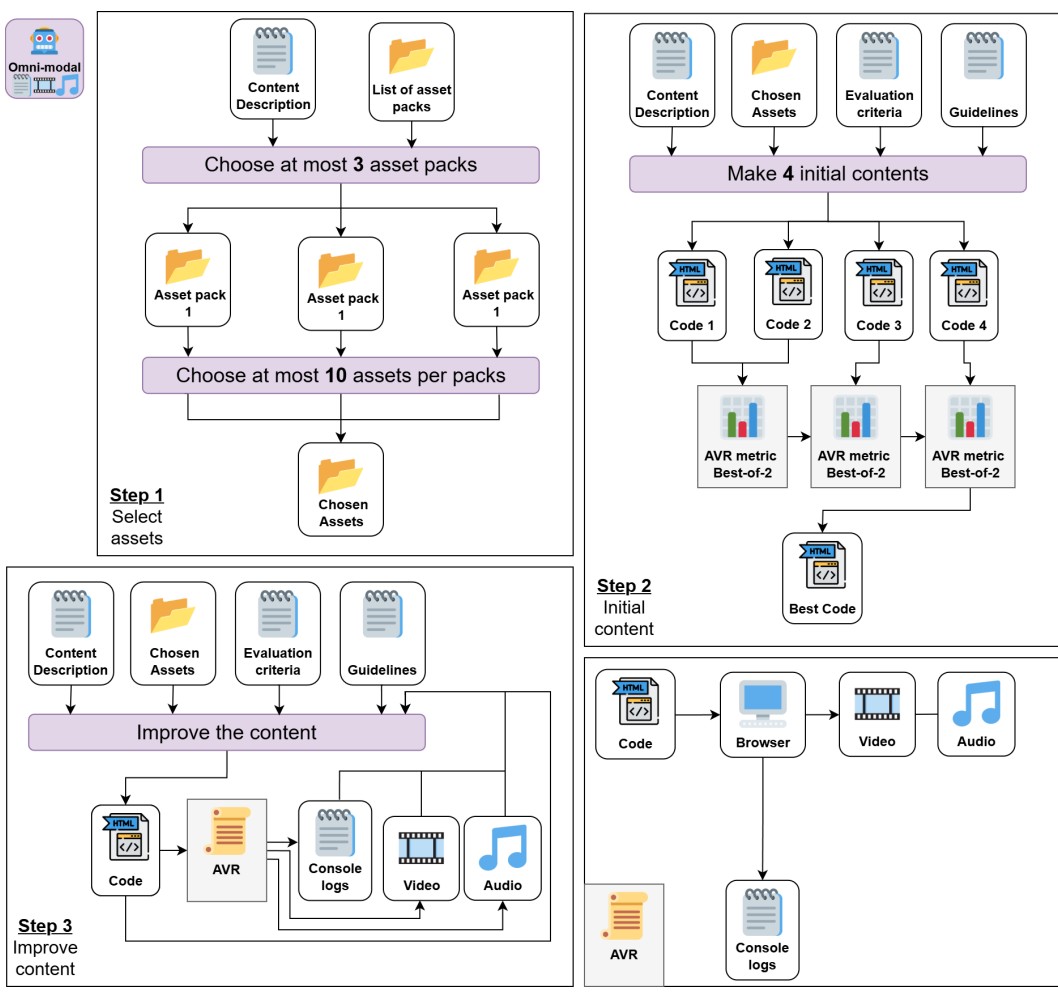

Figure 5: AVR-Agent 2.0: Omni-agent framework for audio-visual content generation

# E  COMPARING AVR-AGENT ON DIFFERENT GAMES AND SETTINGS

We show examples of inital content vs AVR-Agent for game generation with Kimi-K2.

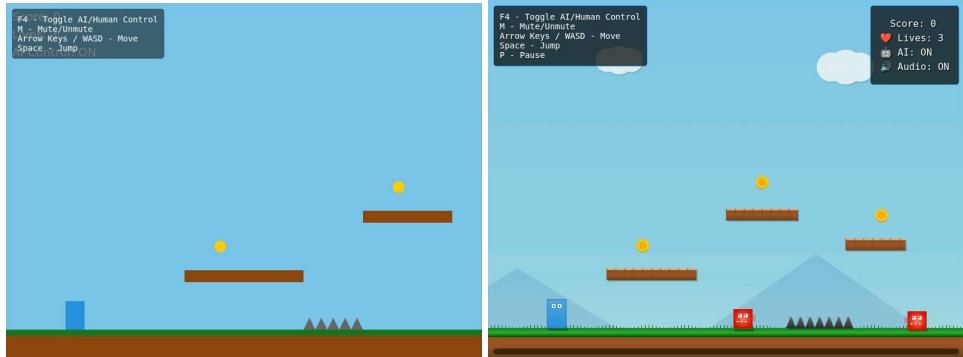

Figure 6: Platformer game generation with Kimi-K2: single prompt vs AVR-Agent (10 iterations) using assets and feedback.

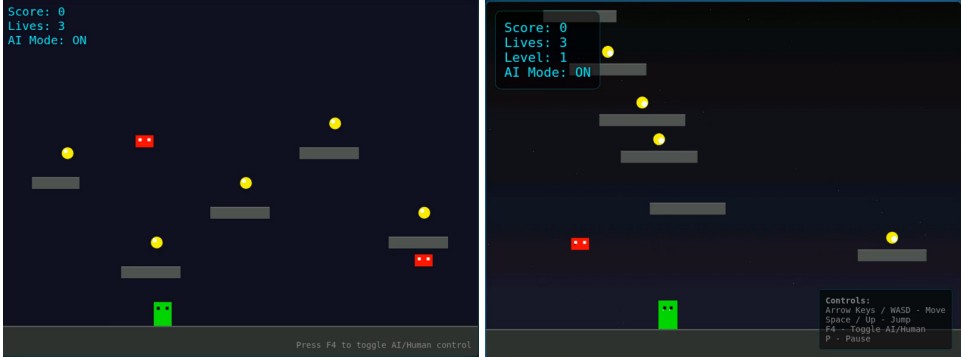

Figure 7: Platformer game generation with Kimi-K2: single prompt vs AVR-Agent (10 iterations) without assets or feedback.

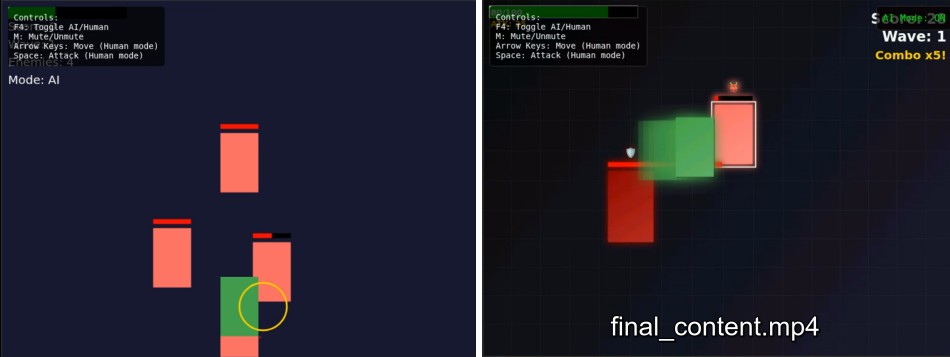

Figure 8: Beat 'em up game generation with Kimi-K2: single prompt vs AVR-Agent (10 iterations) using assets and feedback.

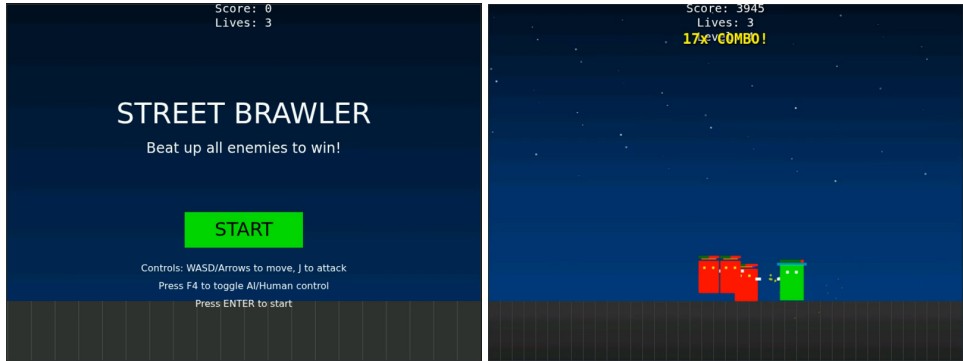

Figure 9: Beat 'em up game generation with Kimi-K2: single prompt vs AVR-Agent (10 iterations) without assets or feedback.

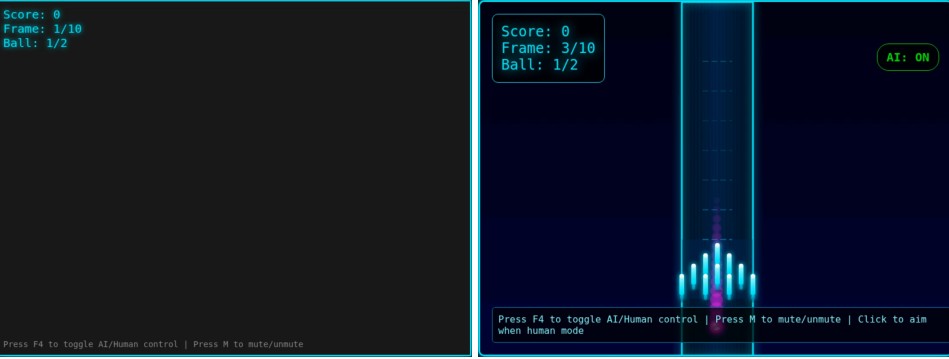

Figure 10: Bowling game generation with Kimi-K2: single prompt vs AVR-Agent (10 iterations) using assets and feedback.

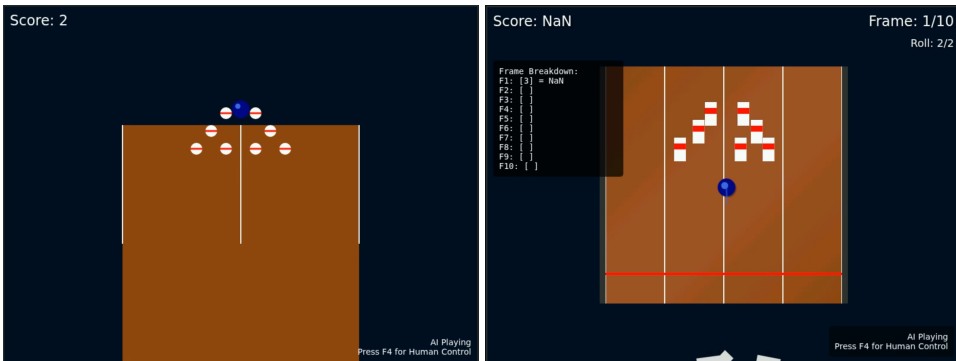

Figure 11: Bowling game generation with Kimi-K2: single prompt vs AVR-Agent (10 iterations) without assets or feedback.

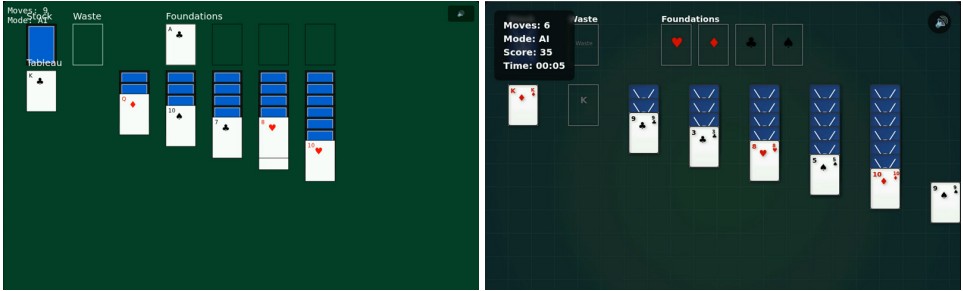

Figure 12: Solitaire game generation with Kimi-K2: single prompt vs AVR-Agent (10 iterations) using assets and feedback.

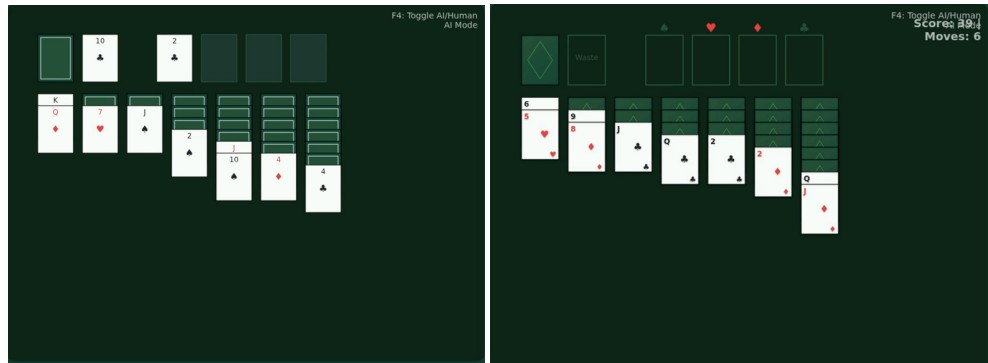

Figure 13: Solitaire game generation with Kimi-K2: single prompt vs AVR-Agent (10 iterations) without assets or feedback.

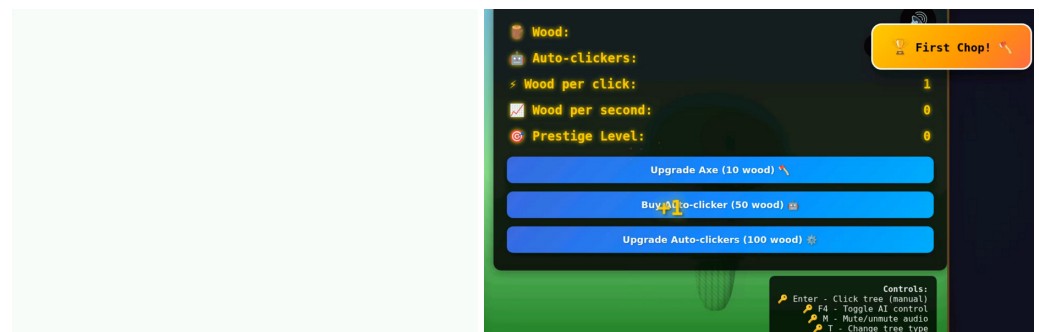

Figure 14: Incremental game generation with Kimi-K2: single prompt vs AVR-Agent (10 iterations) using assets and feedback.

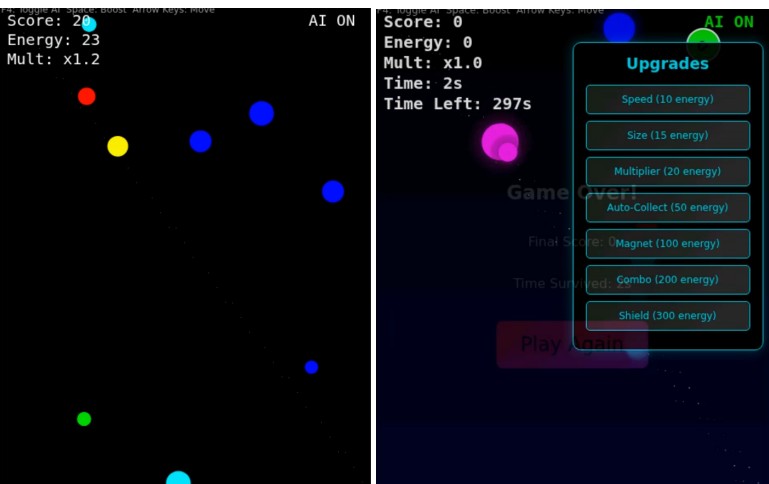

Figure 15: Incremental game generation with Kimi-K2: single prompt vs AVR-Agent (10 iterations) without assets or feedback.

