# OpenReview forum: "Multi-Agent Game Generation and Evaluation via Audio-Visual Recordings"
_ICLR.cc/2026/Conference — Submitted to ICLR 2026_

### Official Review · Reviewer_7AWV · 2025-10-31

**Soundness:** 3
**Presentation:** 3
**Contribution:** 3
**Rating:** 6
**Confidence:** 3

**Summary:**

This paper introduces AVR-Eval, a new evaluation metric for multimedia and interactive content, and AVR-Agent, a multi-agent system to generate JavaScript code from a bank of multimedia assets (audio, images, 3D models) and using AVR feedback. Experiments across 5 games and 5 animations demonstrate that AVR-Agent improves generation quality compared to one-shot LLM generation

**Strengths:**

1.	The presentation is clear and easy to follow.

2.	The paper bridges game generation, multimodal evaluation, and agent-based code improvement, which is a novel combination not widely explored.

3.	The experiments and visualizations are reasonable and well done.

4.	The proposed pairwise audio-visual comparison metric is a meaningful step toward automated evaluation of multimedia and interactive content.

**Weaknesses:**

1.	AVR-Agent primarily combines existing LLMs and evaluation loops rather than introducing fundamentally new generation algorithms or architectures.

2.	The AVR-Eval metric was tested only on simple 2D JavaScript games and animations; its generalizability to complex, commercial-quality or 3D content is unclear.

3.	While open-sourced, the framework depends on large proprietary models (Gemini, Grok, etc.) and external asset libraries, which may limit full replication.

4.	The iterative AVR-Agent pipeline (multiple generations, AV recordings, model feedback) appears computationally expensive, while runtime and cost analyses are missing.

5.	AVR-Eval lacks direct validation against human preference scores.

**Questions:**

1.	What is the time and computational cost of one full AVR-Agent generation cycle (including k-initial candidates and multi-round improvements)?

2.	Can AVR-Agent be applied to generate larger or multi-level games beyond short demos? How does performance scale with content complexity?

3.	Given reliance on commercial APIs and proprietary LLMs, how can other researchers replicate the reported results or extend AVR-Agent using open-source models?

4.	To what extent is the framework autonomous? Does it still require human intervention for debugging or asset curation?

5.	Are the two agents (coding and omni-modal) strictly sequential, or can they interact iteratively (e.g., co-reflection or negotiation)?

---

### Official Review · Reviewer_bTx2 · 2025-11-01

**Soundness:** 2
**Presentation:** 1
**Contribution:** 2
**Rating:** 2
**Confidence:** 2

**Summary:**

The paper aims to address two issues in game generation: lack automated evaluation metrics and struggle with complex content. Specifically, this paper proposes AVR‑Eval, a relative, pairwise metric that compares two pieces of web‑based multimedia (games or animations) using audio‑visual recordings (AVRs) as inputs to an omni‑modal judge, followed by a text‑model review step. An ablation shows that multi‑round description → comparison and a text‑model review notably reduce failure modes. Besides, this paper proposes AVR‑Agent. In the first stage, the coding model selects which assets to use to produce the desired content given
the original description. In the second stage, the coding model is asked to generate the content based on the original description, chosen assets, general guidelines, and evaluation criteria. In the third stage, the content is improved over multiple steps including content description and feedback for the content. Empirical results demonstrate the effectiveness of the Agent and Eval.

**Strengths:**

1. Target an interesting goal in achieving automated game design.
2. Each component in the AVR-Eval or the AVR-Agent is evaluated carefully to demonstrate its effectiveness.

**Weaknesses:**

1. Overall the paper is very engineering for designing pipeline and prompts for AVR-Eval and AVR-Agent, and lack main technical contribution.
2. Not much related work being discussed in the paper so it's hard to place the paper in existing literature.
3. While AVR‑Eval is intuitive and the ablation is convincing, there is no study of alignment with human raters.
4. The benchmark uses five game and animations. Many results may not carry to richer game loops, content pipelines, or larger engine use (e.g., asset streaming, physics edge cases, level generation at scale).
5. The writing should be improved to better separate the discussion about approach and results.

**Questions:**

N/A

---

### Official Review · Reviewer_uKvp · 2025-11-01

**Soundness:** 3
**Presentation:** 2
**Contribution:** 2
**Rating:** 4
**Confidence:** 3

**Summary:**

This paper presents two primary contributions: 1) AVR-Eval, an automated, relative evaluation metric for interactive multimedia content, such as JavaScript games. This metric works by generating audio-visual recordings (AVRs) of two competing pieces of content and feeding them to an omni-modal model (Qwen2.5-Omni-7B) to perform a pairwise comparison. 2) AVR-Agent, a multi-agent framework for generating this JavaScript content. The system uses a text-based coding LLM, guided by feedback from an omni-modal model and console logs, to iteratively refine code. It also has access to a bank of human-made multimedia assets.

**Strengths:**

1. Addresses a Critical Bottleneck: The paper tackles a core challenge in generative AI for interactive content: the lack of scalable, automated evaluation. Human-in-the-loop evaluation (like WebDev Arena) is a major bottleneck, and the idea of using an omni-modal model to "watch and listen" to content is a novel and important research direction.

2. Novelty of the Metric Concept: The AVR-Eval metric moves beyond static code analysis or simple screenshot evaluation. By incorporating audio and video (temporal dynamics), it attempts to capture a more holistic sense of the user experience, which is the correct approach for evaluating games.

3. Interesting (and Honest) Negative Result: The paper's most compelling finding is the failure of the agent to benefit from high-quality assets and AV feedback. This is a valuable, non-trivial result that challenges the community's assumptions about agent capabilities and is worthy of discussion.

**Weaknesses:**

1. Experimental Circularity: The entire experimental setup is critically circular. The AVR-Agent uses an omni-modal model (Qwen2.5-Omni-7B) to provide feedback for improvement. The AVR-Eval metric then uses the exact same model (Qwen2.5-Omni-7B) to judge the final quality. The agent is, therefore, being optimized to satisfy the biases of its own evaluator. The paper does not demonstrate that the agent is producing objectively better games; it only demonstrates that it is getting better at pleasing the Qwen-Omni model. This is a fundamental conflict of interest that invalidates the main results.

2. The AVR-Agent is not a complex multi-agent system. It is a two-model pipeline: a coding LLM that writes code and an omni-modal model that provides text descriptions. This is a standard tool-augmented agent or a simple feedback loop, not a "multi-agent system" in the sense of complex roles, negotiation, or collaborative planning.

3. Misleading Interpretation of the "Gap": The paper claims to have found a "gap between humans and AI" because the agent does not benefit from assets or AV feedback. This is a major misinterpretation of a flaw in their own agent's design.

4. Asset Flaw: The agent does not see or hear the assets. It only selects them based on text metadata (e.g., filenames, dimensions, BPM). A human looks at the art and listens to the music. The agent is choosing blind. It is no surprise it cannot leverage assets it has no direct perceptual access to. This is an agent design flaw, not a fundamental AI limitation.

5. Feedback Flaw: The "audio-visual feedback" is just a high-level text description from the omni-model (e.g., "describe the content," "provide subjective feedback"). This feedback is abstract and non-actionable. How is a coding model supposed to translate "the visual design is not harmonious" into a specific JavaScript code change? The technical challenge is bridging this modality-to-code gap, and the paper's solution (just passing text) fails.

6. No Human Correlation for AVR-Eval: The paper proposes AVR-Eval as an automated substitute for human evaluation. To validate such a metric, it is essential to conduct a human correlation study. The authors must show that AVR-Eval's pairwise preferences (A vs. B) strongly correlate with the preferences of human evaluators on the same content pairs. The paper provides zero such data.

7. Weak Metric Validation: The validation in Table 1 is unconvincing. Detecting "broken" (crash/black screen) or "mislabeled" (fireworks vs. bouncing ball) content is a trivial bar for any modern multimodal model. The only subjective test is against "human-made" content, where the generated content "won" 32.22% of the time. This 1/3 failure rate to identify high-quality human content is deeply concerning and suggests the metric is not a reliable judge of "quality."

8. Benchmark is a Toy Problem: The benchmark of 5 simple animations and 5 simple games (e.g., "Bouncing Ball," "Solitaire") is a toy problem. These tasks do not involve complex game logic, state management, or novel mechanics. It is impossible to generalize any findings from this benchmark to the "challenging problem" of novel video game generation.

9. Inconsistent Experimental Protocol: The authors admit that due to API costs ("paid out-of-pocket"), stronger models were run with fewer iterations than weaker models. This inconsistent protocol makes the model-to-model comparisons in Figure 4c and Table 5 unreliable, as the results are confounded by the different iteration counts.

**Questions:**

1. On Metric Validity: The central claim of this paper is the utility of AVR-Eval. Why did you not conduct a human correlation study to prove that your metric's judgments align with human preferences? Without this, how can we trust any of the results that depend on this metric?

2. On Circularity: How do you defend the experimental design where the agent's feedback mechanism (Qwen2.5-Omni-7B) is the same model as the final evaluator (Qwen2.5-Omni-7B)? How can you demonstrate that your AVR-Agent is learning to make better games and not just learning to game the judge?

3. On the "Asset Gap": You claim to have found a "gap" because the AI does not benefit from assets. Since your agent only sees text metadata and not the visual or audio content of the assets, is this not a simple failure of your agent's design rather than a fundamental finding about AI?

4. On the "Feedback Gap": You claim the agent does not benefit from AV feedback. Your omni-model provides only high-level, non-actionable text descriptions. Did you attempt to provide more structured, code-oriented feedback (e.g., "The enemy hitbox at (x,y) is too large," "The jump sound is not triggering on line 87")?

5. On Generalizability: Given that your benchmark is limited to 10 extremely simple, well-defined problems, how can you be confident that your findings, especially the surprising lack of benefit from assets and feedback, will generalize to the generation of even moderately complex, novel video games?

---

### Official Review · Reviewer_Dbqv · 2025-11-16

**Soundness:** 2
**Presentation:** 1
**Contribution:** 2
**Rating:** 4
**Confidence:** 4

**Summary:**

The paper focuses on the direction of web-based coding agents and introduces AVR-Eval, a new automated metric for evaluating the quality of multimedia content such as web games and animations. Furthermore, the authors also proposes AVR-Agent, a multi-agent framework that generates JavaScript code by leveraging a multimedia asset library and iteratively refines outputs through AVR feedback. Experimental results on several game and animation tasks demonstrate the effectiveness of both AVR-Eval and AVR-Agent.

**Strengths:**

- The paper first proposes a new evaluation metric AVR-Eval that uses audio-visual recordings and omni-modal reasoning models to assess multimedia content.

    - AVR-Agent introduces a well-designed multi-agent framework for JavaScript-based multimedia content generation by leveraging a bank of multimedia assets.

    - The combination of AVR-Eval and AVR-Agent provides an effective framework for establishing a closed loop of generation, evaluation, and refinement.

**Weaknesses:**

- The benchmark contains only 10 simple tasks (5 animations, 5 games), which are insufficient to support broad claims about model performance or generalization. The proposed method can be further evaluated on complex 3D games and long-term interactive tasks to verify its effectiveness.

    - The authors only discussed why FVD  is unsuitable in the introduction section without providing any comparative experiments. In addition, the paper does not compare AVR-Eval with other common evaluation metrics (e.g., CLIPScore [1], GPTScore [2]) on the same content. Therefore, it is recommended that the authors further validate the performance of AVR-Eval by comparing it with traditional evaluation metrics (e.g., CLIPScore, GPTScore).

    - It remains unclear whether AVR-Eval aligns well with human preferences or only performs well on some specifical tasks.

    - The design of AVR-Eval is primarily heuristic and empirically motivated, without any theoretical analysis. It is recommended that the authors add a more systematic and theoretically grounded explanation.

    - The writing of this paper can be further improved to enhance its overall clarity, for example, by presenting the Related Work section as an independent part.

    [1] Hessel J, Holtzman A, Forbes M, et al. CLIPScore: A Reference-free Evaluation Metric for Image Captioning[C]//EMNLP (1). 2021.
    [2] Fu J, Ng S K, Jiang Z, et al. GPTScore: Evaluate as You Desire[C]//Proceedings of the 2024 Conference of the North American Chapter of the Association for Computational Linguistics: Human Language Technologies (Volume 1: Long Papers). 2024: 6556-6576.

**Questions:**

Please refer to the Weaknesses.

---

### Meta-Review · Area_Chair_gGiF · 2026-01-05

**Summary:**

The overall feedback for this paper was negative, mainly highlighting simplicity of the benchmark, unclear or misleading writing, lack of human alignment studies, and inconsistent not-reproducible evaluation protocol.

**Reviewer Concerns:**

The was no rebuttal.

**Reviewer Scores:**

No change.

---

### Decision · Program_Chairs · 2026-01-26

Reject